# Bioleaching of Pyrrhotite with Bacterial Adaptation and Biological Oxidation for Iron Recovery

**Bong-Ju Kim, Yong-Kwon Koh and Jang-Soon Kwon ***

Korea Atomic Energy Research Institute, Deadeok-daero 989, Yuseong-Gu, Daejeon 34057, Korea;
kbj7878@kaeri.re.kr (B.-J.K.); nykkoh@kaeri.re.kr (Y.-K.K.)
* Correspondence: jskwon@kaeri.re.kr; Tel.: +82-42-868-2904

**Abstract:** The microbially mediated recovery of valuable metals contained in mining waste presents an economical alternative to conventional hydrometallurgical processes. In order to investigate the effect of bacterial adaptation and biological oxidation on bioleaching, the microbially mediated bioleaching of a pyrrhotite sample from mine waste, with indigenous bacteria existing in acid mine drainage, was studied. The indigenous bacteria were sub-cultured repeatedly for iron adaptation, and *Acidithiobacillus ferrooxidans* was identified as the dominant member of the microbial consortium. The point of zero charge (PZC) of pyrrhotite sampled from mine waste was determined as 3.0. The performance of bioleaching by contact and non-contact biological oxidation was compared by conducting bioleaching under different initial pH ($pH_{ini}$) conditions (2.8 and 3.2). Negatively charged bacteria could be attached onto the pyrrhotite, which has a positive surface charge at lower $pH_{ini}$ (2.8) than the PZC (3.0). Bacteria attachment and corrosion pits on the surface of the pyrrhotite residues were observed at $pH_{ini}$ of 2.8. Under bacteria-adapted conditions, the leaching concentration of Fe (44.2 mg/L) at $pH_{ini}$ of 2.8 was 2.1 times greater than that (21.3 mg/L) at $pH_{ini}$ of 3.2. Under non-adapted bacteria conditions, the extent of Fe leaching was not significantly different between the $pH_{ini}$ of 2.8 and 3.2. This could be attributed to the fact that the adapted bacteria could more easily attach onto the pyrrhotite surfaces at $pH_{ini}$ 2.8, allowing contact biological oxidation during the bioleaching experiments. We demonstrate here that the bioleaching of pyrrhotite could increase Fe recovery through bacterial adaptation and contact biological oxidation.

**Keywords:** iron recovery; pyrrhotite; bioleaching; point of zero charge; bacterial adaptation; contact biological oxidation

## 1. Introduction

Bioleaching is a technique used to recover valuable metals from sulfide minerals. Sulfide minerals are sources of valuable metals such as Fe, Pb, Zn, Cd, and Cu, and the valuable metals are available through the separation and decomposition of sulfide minerals. Traditionally, physicochemical treatment methods such as roasting, flotation, and acid or base leaching have been used to decompose sulfide minerals. Currently, the bioleaching method is attracting interest as an alternative method due to its environmental friendliness and cost effectiveness.

Ever since it became a known fact that acidophilic bacteria can survive in a strongly acidic environment [1], bioleaching has been widely used for the recovery of valuable metals from slime or low-grade ore minerals and the removal of heavy metals from polluted soils. Various studies using the bioleaching technique to recover valuable metals (Fe, Cu, Pb, Zn, etc.) from sulfide minerals including chalcopyrite [2], arsenopyrite [3,4], molybdenite [5,6], galena [7–9], and sphalerite [9,10] have been reported. In addition, indigenous acidophilic bacteria have been already used for the recovery of Cu, Au, and U in slime [11–13].

The bioleaching of sulfide minerals for metal recovery can be regarded as a type of bio-oxidation that can be classified as contact oxidation or non-contact oxidation [14,15].

Contact oxidation means oxidation by microbes that are placed onto the surface of the sulfide mineral, while non-contact oxidation means the oxidation of sulfides by ferric ion ($Fe^{3+}$), which occurs through the microbial catalytic oxidation of ferrous ion. In the case of contact oxidation, bacteria are able to select sites on which they can easily obtain energy through oxidization [16]. Generally, these sites are concentrated on the surfaces of sulfide minerals where they are characterized as mechanically and chemically weak crystallization, dislocation, and corrosion pits [17–19].

In the non-contact mechanism, the bacterial oxidation of ferrous ion ($Fe^{2+}$) to ferric ion ($Fe^{3+}$) regenerates the oxidant necessary for further reaction with the mineral. This mechanism has been shown to proceed via two major routes: the thiosulfate and polysulfide pathways [17–19]. Acid-insoluble sulfides such as $FeS_2$, $MoS_2$, and $WS_2$ are non-contact bioleaching by the ferric ions, generating thiosulfate, which is further oxidized to form sulfuric acid. In contrast, the dissolution of a soluble metal sulfide involves a proton attacking the mineral surface.

Bioleaching might be also affected by various factors including bacterial strain, pH, oxidation/reduction potential (ORP), temperature, and heavy metals in leachate. During the bioleaching processes, bacterial viability can be especially affected by toxic heavy metal ions, resulting in decreased recovery of the available elements from sulfide minerals. Therefore, bacterial adaptation to toxic heavy metals can play a significant role in enhancing the efficiency of bioleaching [20,21].

The aim of this study was to investigate the effects of bacterial adaptation and biological oxidation on the bioleaching of pyrrhotite using an indigenous bacterium. Bacterial adaptation was performed to examine the effect of the adaptation of indigenous bacteria. Bioleaching experiments were conducted to estimate the effect contact biological oxidation on bioleaching efficiency at different pHs using non-adapted and adapted indigenous bacteria.

## 2. Materials and Methods

### 2.1. Mine Waste and Pyrrhotite

In order to prepare the pyrrhotite sample used in the experiment, mine waste was acquired from an abandoned iron mine located in Ul-Jin, Korea. In the laboratory, the mine waste was crushed using a jaw crusher and a cone crusher, ground, and then sieved with a 20 mesh (841 micron, ASTM) sieve to exclude accessory minerals, quartz especially. The sieved mine waste sample was separated into magnetic and non-magnetic fractions under the magnetic field intensity of 0.2 T using a magnetic separator (Frantz magnetic separator, L-ICN, Frantz, Sterling, IL, USA) due to the highly magnetic properties of pyrrhotite. The effect of the magnetic separation process on the magnetic properties was evaluated using a vibration sample magnetometer (VSM, BHV-50HTI, Riken, Keiki, Tokyo, Japan). The chemical composition of the mine waste was analyzed using an atomic absorption spectrophotometer (AAS, AA-7000, Shimadzu, Kyoto, Japan) through aqua regia digestion. The powdered form of the pyrrhotite sample magnetic fraction was characterized using a powder X-ray diffractometer (XRD, D8ADVANCE, Bruker, Billerica, MA, USA) with Cu-Kα radiation (1.5406 Å) at a scanning speed of 2°/min. The point of zero charge (PZC) of the pyrrhotite sample was determined by measuring the zeta potential (Zetasizer, Nano-ZS MPT-2, Malvern, PA, USA) at various pH values, which were controlled using $HNO_3$ and $NaOH$.

### 2.2. Microbes

2.2.1. Collection and Cultivation of Microbes

The indigenous microbes used in this bioleaching experiment were collected from the mine drainage (pH: 4.62 and ORP: 365 mV) located 82 m under the ground at the Ul-Jin Mine cave. The microbes were cultured in a mineral-salt medium (MSM). The MSM was prepared by dissolving 0.2 g $(NH_4)_2SO_4$, 0.5 g $MgSO_4·7H_2O$, 0.25 g $CaCl_2$, 3.0 g $KH_2PO_4$, and 0.05 g $FeSO_4$ in 1.0 L of distilled water [10]. The growth medium was prepared by adding elemental sulfur with a concentration of 1.0 g/L to the MSM. After 7 days of

incubation at 32 °C, the enrichment cultures were plated onto MSM agar supplemented with 0.5 wt % $FeSO_4$ to isolate the iron-resistant bacteria. Of the colonies isolated, a single colony was used for bioleaching investigation.

### 2.2.2. Identification of the Bacterial Strain

The 16S rRNA gene of the iron-resistant bacteria was amplified by polymerase chain reaction using a HotStarTaq[®®] Plus Master Mix Kit (Qiagen, Valencia, CA, USA), with the universal primers 27F (5′-AGAGTTTGATCMTGGCTCAG-3′) and 1492R (5′-GGYTACCTTGTTACGAC-TT-3′), which were sequenced at Macrogen Corp. (Daejeon, Korea). The sequenced gene was compared with the sequences available in the GenBank databases using the BlastN program from the National Center for Biotechnology Information homepage.

### 2.3. Adaption of Iron-Resistant Bacteria

The iron-resistant bacteria from the bacterial culture were enriched by culturing in the MSM with 1% $FeSO_4$ for 21 days, and this culturing was repeated several times as necessary to enhance the iron resistivity (adaptation). Then, we adapted the enriched bacteria to iron ions using the same adaptation medium (MSM with 1% iron) [13]. During enrichment and adaptation, the pH and ORP in the media were measured using a Thermo Scientific Orion 3Star portable meter (Orion 3Star Portable, Thermo Scientific, Beverly, MA, USA) to confirm the ability of the bacteria to oxidize the pyrrhotite sample. This meter was equipped with a pH probe (Triode gel-filled epoxy-body LM, 9107WMMD) and ORP electrode (Sure-flow combination redox/ORP electrode, 9678BNWP, Ag/AgCl), and all electrodes were calibrated daily using standard procedures during the experiments.

### 2.4. Bioleaching Experiments

Bioleaching experiments were conducted to evaluate the effect of bacterial adaptation on the bioleaching efficiency. Batch-type bioleaching experiments were prepared in 500-mL Erlenmeyer flasks containing 300 mL of culture medium (290 mL of fresh MSM + 10 mL of adapted bacteria medium) amended with 3 g of 20 mesh sieved pyrrhotite sample particles. A control experiment was also carried out simultaneously under abiotic conditions to check the contribution of simple chemical leaching from the growth medium. After the leaching experiments, the iron (Fe) concentration in the leachates was analyzed using an atomic absorption spectrophotometer (AAS). The morphological features of the residues were observed using a scanning electron microscopy (SEM, S4800, Hitachi, Tokyo, Japan). The mass of Fe leached per unit mass of the pyrrhotite sample was applied to the following Equation (1):

$$E = E_I \left(1 - e^{-kt}\right) \tag{1}$$

where $E$ is the leaching concentration at time $t$, $E_I$ is the maximum leaching concentration, and $k$ is the leaching rate constant.

## 3. Results and Discussion

### 3.1. Pyrrhotite in the Mine Waste

The chemical analysis shows that the mine waste of 1 kg in mass obtained from the abandoned Ul-Jin Fe mine contained 5670 mg of Fe, 140 mg of Pb, 479 mg of Cu, and 238 mg of Zn.

The XRD result of two classified fractions after magnetic separation of the mine waste (Figure 1) demonstrated that the magnetic fraction is composed of pyrrhotite with quartz as a minor impurity, whereas the non-magnetic fraction includes various minerals such as galena, pyrites, pyrrhotite, quartz, and sphalerite.

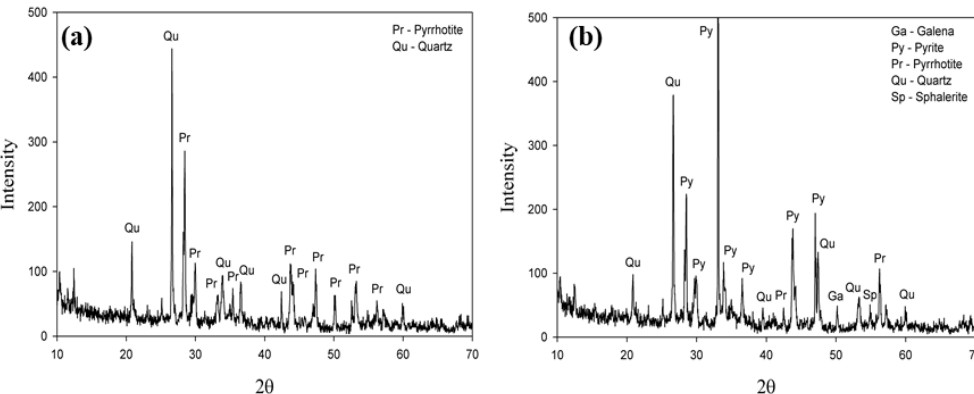

**Figure 1.** XRD results of the mine waste after magnetic separation: (**a**) magnetic fraction; (**b**) non-magnetic fraction.

The distribution of magnetization along the mineral composition is illustrated by comparing the magnetic hysteresis loops of the magnetic and non-magnetic fractions in the mine waste (Figure 2). The maximized magnetization of the two fractions was 3.1 and 69.8 $Am^2/kg$ in the non-magnetic and magnetic fractions, respectively. In addition, in the case of magnetically separated samples, magnetic minerals have ferromagnetic properties, and non-magnetic minerals have no magnetism.

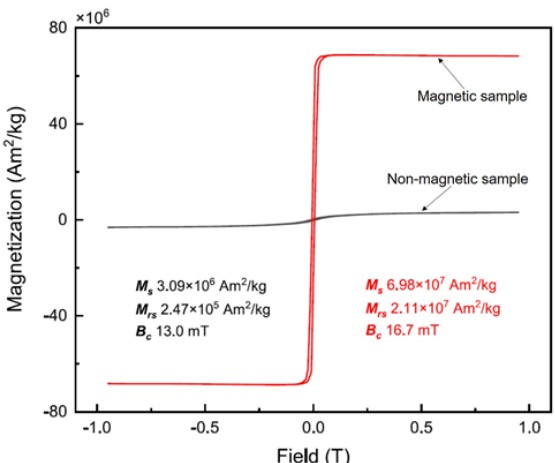

**Figure 2.** Magnetic hysteresis loops of (non-) magnetic fractions in the mine waste sample.

The zeta potential of the pyrrhotite sample (magnetic fraction) measured under various pH conditions is shown in Figure 3. As the pH increased from 1.0 to 9.0, the zeta potential of the pyrrhotite decreased from +8.1 to −25.6 mV. The $pH_{pzc}$ value of the pyrrhotite was determined to be 3.0, which is similar to the result of Widler and Seward [22], who reported that the $pH_{pzc}$ of artificial pyrrhotite produced at 230 °C and 100 atm was 2.7. This indicates that the pyrrhotite collected from the mine waste was effectively separated for the purpose of these experiments. It is obvious that solid material is positively charged when the pH is lower than the $pH_{pzc}$, while it is negatively charged when the pH is higher than the $pH_{pzc}$.

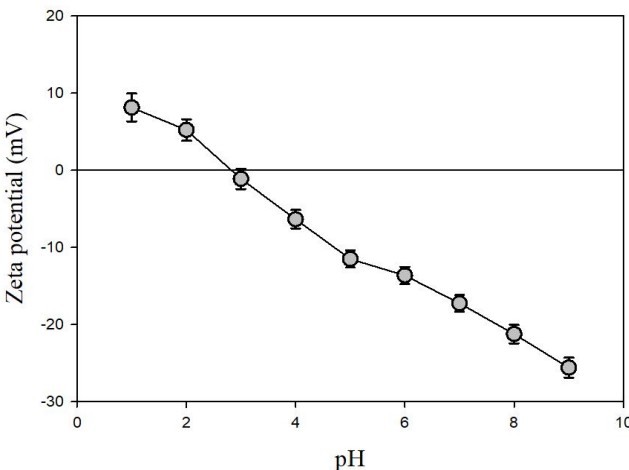

**Figure 3.** Zeta potentials of pyrrhotite used in the experiments.

### 3.2. Bacterial Adaptation

The results of the phylogenetic analysis show that *A. ferrooxidans* is the dominant member among the four-step adapted bacteria (Figure 4). *A. ferrooxidans*, known as an Fe- and S-oxidizing acidophile, has a well characterized genetic system and has been previously shown to be a ubiquitous member of the acid mine drainage ecosystem [23]. The presence of *A. thiooxidans* and *A. thermosulfidooxidans* also indicates the potential for thermophilic enrichment [24].

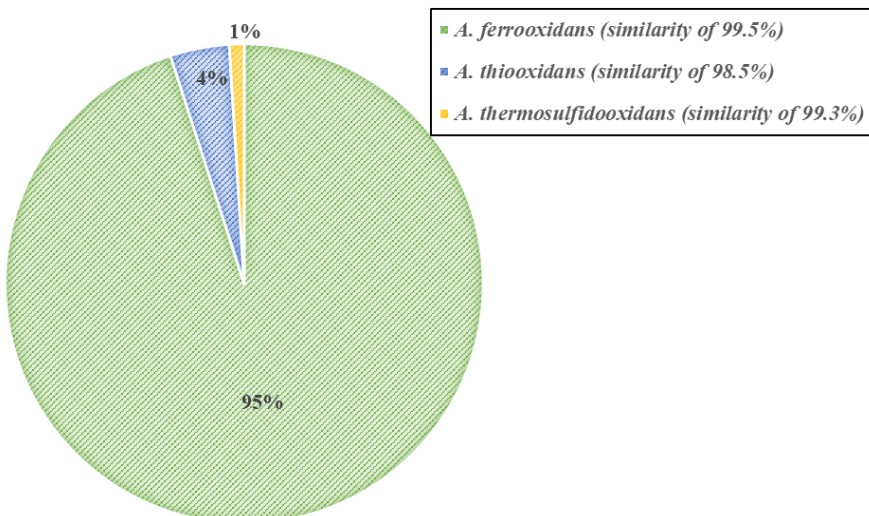

**Figure 4.** Microbial profiling of the four-step adapted bacteria.

The adaptation of *A. ferrooxidans* was performed in four-step adaptation cycles, based on the report of Tuovinen, Niemelä, and Gyllenberg [24]. The variations of pH and ORP in the culture medium during the adaptation cycles are presented in Figure 5.

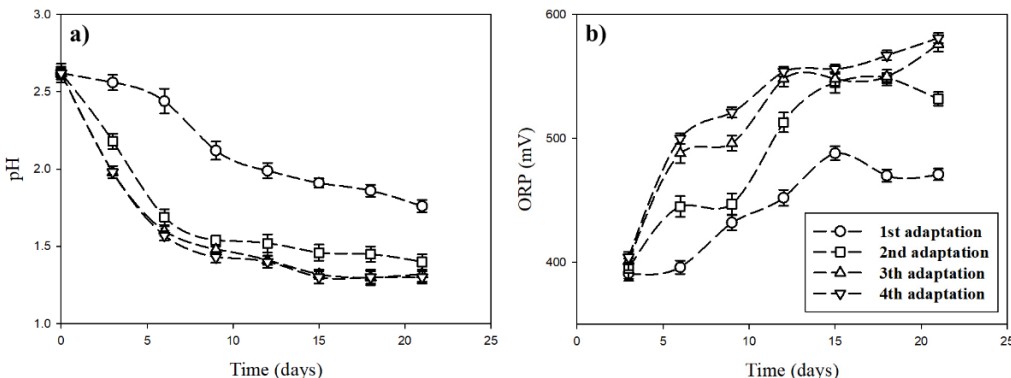

**Figure 5.** Variations of pH (**a**) and oxidation/reduction potential (ORP) (**b**) in the culture medium during different adaptation cycles.

The estimated results show that the pH in the culture medium decreased with the increase of adaptation time, whereas ORP increased with time. This might be due to the biological oxidation of energy sources ($Fe^{2+}$) by *A. ferrooxidans* along with the inorganic oxidation of oxidative energy sources ($Fe^{2+}$) partially during the adaptation. Differences in pH decreases and ORP increases in adaptation cycles were observed. In the first adaptation, pH decreased from 2.6 to 1.8, whereas ORP increased from 390 to 470 mV. In the fourth adaptation, pH decreased, starting from 2.6 to 1.3 after 21 days, whereas ORP increased from 400 to 580 mV. In the third and fourth adaptations, there was no significant difference in the results of pH and ORP estimated with the adaptation time. It was expected that *A. ferrooxidans* sufficiently adapted to iron ions at least by the third adaptation. Moreover, in the third and fourth adaptations, the pH showed an equilibrium constant after 15 days of inoculation, which indicates that the optimum adaptation of *A. ferrooxidans* was reached by day 15. Previous studies have reported optimum adaptation periods of one week [25,26], two weeks [27–29], and four weeks [30,31]. Reference [20] also mentioned that the repetition of adaptation performance leads to a reduction in the lag phase of bacteria and improved biological oxidation capacity of bacteria.

*3.3. pH and ORP Changes through Pyrrhotite Bioleaching*

The variations of pH and ORP during bioleaching of the pyrrhotite under different initial pH conditions ($pH_{ini}$ 2.8 and 3.2) are presented in Figure 6. At the initial $pH_{ini}$ of 2.8, the pH remained relatively constant regardless of bioleaching time in bacterial adaptation conditions, although a slight pH fluctuation was observed at 41 days (Figure 6a). pH increased gradually with time up to 3.9 under the abiotic condition and up to 4.1 in the non-adapted bacterial condition. The ORP values decreased in all three experimental conditions, and the variances were intensive in the order of adaptation < non-adaptation < abiotic conditions (Figure 6b). The reason for the opposite pH and ORP changes during the oxidation of pyrrhotite could be explained by the inorganic/biological oxidation process occurring as described in Reactions (2) and (3).

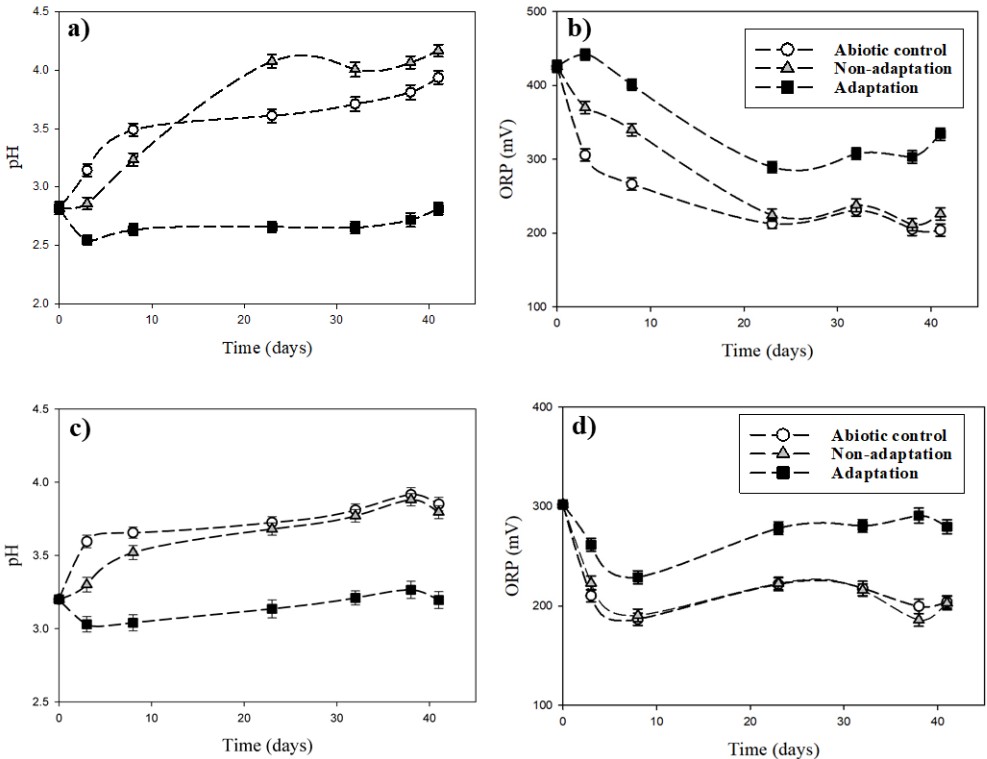

**Figure 6.** Variations of pH and ORP during bioleaching of the pyrrhotite under different initial pH conditions: (**a**) and (**b**), $pH_{ini.}$ 2.8; (**c**) and (**d**) $pH_{ini.}$ 3.2.

The pH values in the non-adaptation condition were slightly lower than those in the abiotic condition (Figure 6c). The ORP values decreased up to day 10 of the experiments and then fluctuated thereafter in all three experimental conditions. The ORP values were in the order of adaptation > non-adaptation ≈ abiotic condition (Figure 6d). The pH increase in the abiotic condition was affected by hydrogen ions being consumed by the chemical oxidation of pyrrhotite. The decrease of ORP and increase of pH are caused by the chemical characteristics of pyrrhotite in dissolution [32,33]. Dissolution reactions of pyrrhotite can be expressed by Reactions (2) and (3) [34]:

$$Fe_{1-x}S + 2H^+ \rightarrow (1-3x)Fe^{2+} + 2xFe^{3+} + H_2S \tag{2}$$

$$2Fe_{1-x}S + O_2 + 4H^+ \rightarrow (2-6x)Fe^{2+} + 4xFe^{3+} + 2S^0 + 2H_2O. \tag{3}$$

Reaction (2) represents the non-oxidative dissolution of pyrrhotite. Reaction (3) indicates the oxidative dissolution of pyrrhotite in the presence of oxygen. Although indigenous bacteria were present in the non-adaptation condition, the same patterns of pH and ORP changes for the abiotic condition were observed. This indicates that indigenous bacteria do not have a major role in biological oxidation in the non-adaptation condition, but chemical oxidation could contribute to changes of pH and ORP. Biological oxidation by indigenous bacteria using pyrrhotite S and Fe ions can be expressed by Reactions (4) and (5):

$$S^0 + 1.5O_2 + H_2O \rightarrow SO_4^{2-} + 2H^+ \tag{4}$$

$$4Fe^{2+} + O_2 + 4H^+ \rightarrow 4Fe^{3+} + 2H_2O. \tag{5}$$

As mentioned above, the values of pH in the case of the adaptation condition were lower than those in the abiotic and non-adaptation conditions, whereas the ORP values in the adaptation condition were higher than those in the abiotic and non-adaptation conditions. This could be ascribed to biological oxidation by active adapted indigenous bacteria in the adaptation condition, resulting in the discontinuation of pH increase and

ORP decrease caused by the chemical reactions (non-oxidation and direct oxidation of pyrrhotite) described in Reactions (2) and (3).

Biological attachment and bio-oxidation of the indigenous acidophilic adapted bacteria onto the pyrrhotite residues after the bioleaching under different $pH_{ini}$ conditions was presented in Figure 7. The rod-shaped bacteria were highly attached onto the residue under the $pH_{ini}$ 2.8 compared to the $pH_{ini}$ 3.2. Moreover, corrosion pits by the adapted bacteria were observed on the surface of pyrrhotite residue under the $pH_{ini}$ 2.8 (Figure 7a). Under the acidic condition, the negatively charged bacteria could be easily attached at the positively charged places of minerals, which are characterized by the crystallographic imperfection [17–19]. The measurement of zeta-potential of the adapted bacteria would have been helpful to interpret the electrostatic attachment between the pyrrhotite and the *A. ferrooxidans,* but the measurement was not performed in this study. Nevertheless, several previous studies [15,19,34] have reported that the $pHs_{pzc}$ of the *A. ferrooxidans* are about 2.0, and these results could support the electrostatic interaction between the *A. ferrooxidans* and the pyrrhotite used in this study under the acidic condition (less than the $pH_{pzc}$ (3.0) of the pyrrhotite). In addition to the electrostatic attachment, the secretion of slime layer, protein-binding receptors, physical adsorption, and hydrophobic interactions of microbes could be also a function of bacteria attachment onto minerals [14–20].

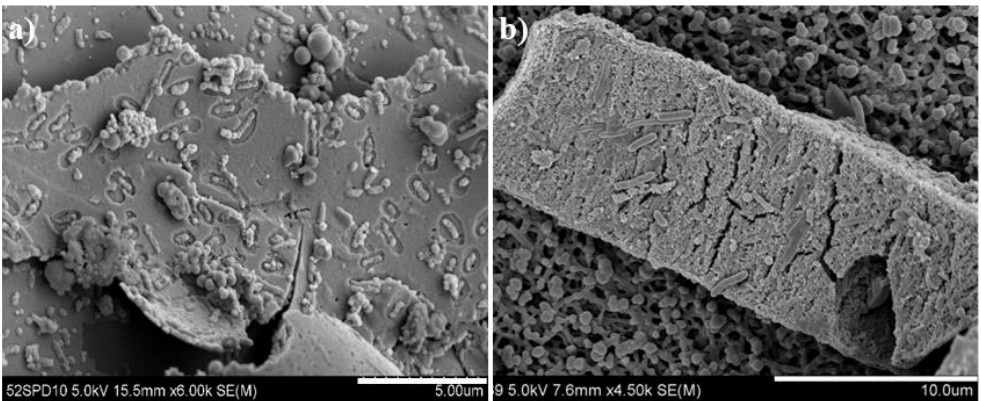

**Figure 7.** SEM images of pyrrhotite residues after completion of bioleaching under different initial pH conditions: (**a**) $pH_{ini}$ 2.8; (**b**) $pH_{ini}$ 3.2.

### 3.4. Fe recovery from Pyrrhotite Bioleaching

The variations of iron (Fe) concentrations in the leachates through bioleaching of the pyrrhotite under different initial pH conditions ($pH_{ini}$ 2.8 and 3.2) are presented in Figure 8. At $pH_{ini}$ 2.8 (Figure 8a), small amounts of iron (Fe) were dissolved throughout the experiment under the abiotic condition and remained low during the experiments, with a concentration of 1.9 mg/L by day 41. Under the non-adaptation condition, 23.8 mg/L of dissolved Fe was leached by day 41. In case of the adaptation condition, dissolved Fe reached 47.4 mg/L by day 41. At the $pH_{ini}$ of 3.2 (Figure 8b), iron (Fe) was rarely dissolved under the abiotic condition, with just 0.7 mg/L at the end of leaching. In the non-adaptation and adaptation conditions, Fe concentrations increased gradually with time, arriving at 19.4 and 21.9 mg/L, respectively, by day 41. This indicated that Fe recovery from the pyrrhotite bioleaching might be increased through the bacterial adaptation of indigenous bacteria.

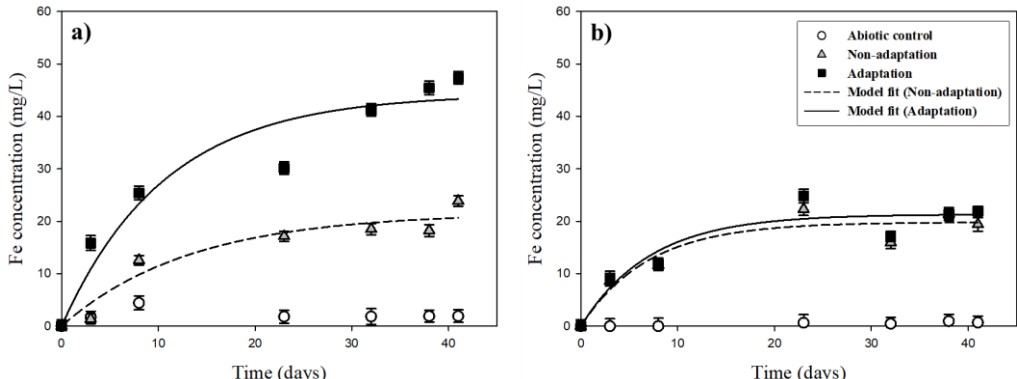

**Figure 8.** Variation of Fe concentrations during bioleaching experiments under different initial pH conditions: (**a**) $pH_{ini}$ 2.8; (**b**) $pH_{ini}$ 3.2.

Elzeky and Attia [27] performed bioleaching experiments using pyrite ($FeS_2$), chalcopyrite ($CuFeS_2$), and arsenopyrite (FeAsS) using non-adapted and adapted bacteria, and they reported a higher performance of metal recovery (3.43 times for Cu and 1.65 times for Fe) under the bacteria adaptation environment than that of non-adaptation. The Fe-adapted microbes demonstrate higher oxidizing power than non-adapted microbes, which resulted from the increased tolerance to the heavy metal [12].

Throughout the bioleaching of the pyrrhotite, the maximum leaching concentration ($E_I$) and leaching rate constant ($k$) were calculated (Table 1). At the $pH_{ini}$ of 2.8, the $E_I$ value (44.2 mg/L) under the bacterial adaptation condition was 2.1 times greater than that (21.6 mg/L) of the non-adaptation condition, whereas the $k$ values were similar. At the $pH_{ini}$ of 3.2, the $E_I$ value and $k$ value were the same or showed little difference. This indicated that bacterial adaptation enhanced the bioleaching of Fe at $pH_{ini}$ 3.0 because the $pH_{pzc}$ of the pyrrhotite used here was positively charged under pH conditions lower than pH 3.0. In addition, previous studies have shown that the $pH_{pzc}$ of *A. ferrooxidans* is in the range of 2.1 to 2.4 [35]. Therefore, it is expected that at the $pH_{ini}$ of 2.8, the negatively charged *A. ferrooxidans* attaches well onto the positively charged surfaces of pyrrhotite due to an electrical attraction; that is, the contact biological oxidation of the pyrrhotite could coincide at the $pH_{ini}$ of 2.8, whereas non-contact biological oxidation is the dominant reaction at $pH_{ini}$ 3.2.

**Table 1.** The maximum leaching concentration ($E_I$) and leaching rate constant ($k$) calculated using Equation (1).

| Experimental Conditions | | $E_I$ (mg/L) | $k$ (1/Day) | $R^2$ |
|---|---|---|---|---|
| $pH_{ini}$ 2.8 | non-adaptation | 21.6 | 0.08 | 0.93 |
| | adaptation | 44.2 | 0.09 | 0.93 |
| $pH_{ini}$ 3.2 | non-adaptation | 19.8 | 0.14 | 0.91 |
| | adaptation | 21.3 | 0.14 | 0.90 |

Xia et al. [20] and Kim et al. [21] conducted bioleaching of pyrites using adapted *A. ferrooxidans* and reported that the adapted bacteria increase the recovery of valuable metals from spent refinery catalysts because the adapted bacteria have increased resistance to the toxicity of heavy metals. They also reported that the adaptation could improve the ability of bacteria to adhere to mineral surfaces, as the adapted bacteria has a lower interaction energy with the surfaces than the non-adapted one.

**4. Conclusions**

Various bioleaching experiments characterized by bio-adaptation and bio-oxidation were carried out using a pyrrhotite sample ($pH_{pzc}$ of 3.0) obtained from an abandoned mine and indigenous bacteria (*A. ferrooxidans*) inhabiting the acid mine drainage in order

to enhance the recovery of valuable metal (iron) from mine waste. During the bioleaching reaction, the leachate pH and ORP under the iron adapted bacteria condition were lower and higher, respectively, than those of the non-adapted and abiotic conditions. Iron recovery was also enhanced under the adapted bacteria condition. The maximum leaching concentration of iron at the $pH_{ini}$ of 2.8 was 44.2 mg/L, which is 2.1 times greater than that at the $pH_{ini}$ of 3.2. Bacteria attachment and corrosion pits on the surface of the pyrrhotite residues were observed at $pH_{ini}$ of 2.8 under the adapted condition. These phenomena could be attributed to the fact that the adapted bacteria could more easily attach to the pyrrhotite surface at pH 2.8; therefore, contact biological oxidation occurred during the bioleaching experiments. Under the non-adapted bacteria condition, iron leaching was not significantly enhanced compared to the abiotic condition. We demonstrated that the bioleaching of pyrrhotite could be enhanced through bacterial adaptation and contact biological oxidation.

**Author Contributions:** B.-J.K. carried out the bioleaching experiments, participated in the reference research, and drafted the manuscript. Y.-K.K. participated in the design of the study and carried out the model analysis. J.-S.K. revised and edited the manuscript. J.-S.K. conceived of the study, participated in its design and coordination, and helped to draft the manuscript. All authors have read and agreed to the published version of the manuscript.

**Funding:** This research was funded by the National Research Foundation of Korea (NRF) funded by the Korean government, Ministry of Science and ICT (No. 2017M2A8A5014859).

**Data Availability Statement:** Data sharing not applicable.

**Acknowledgments:** This work was supported by the National Research Foundation of Korea (NRF) funded by the Korean government, Ministry of Science and ICT (No. 2017M2A8A5014859), and Korea Institute of Energy Technology Evaluation and Planning (KETEP) grant funded by the Korea government ministry of trade, industry and energy (No. 20171510300670, 20193210100120).

**Conflicts of Interest:** The authors declare no conflict of interest.

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
