# Peer review of "Bioleaching of Pyrrhotite with Bacterial Adaptation and Biological Oxidation for Iron Recovery"

_metals, doi:10.3390/met11020295_

Round 1

Reviewer 1 Report

Typos

page 3  line 113

“….the bacterial culture was were enriched by culturing:..”

Line 125

“…of 20 mesh sieved pyrrhotite sample particles (20 mesh sieved).”

Content questions

Page 2 paragraph 2.1

How did you measure the Zeta potential of ore? Do you use any equipment?

Page 5 fig. 4

I am surprised that the consortium consists of 3 strains. I expected a higher variety of microorganisms.

Page 7 paragraph 3.3

Eh develops differently to the Eh of adaptation.  The pH in both experiments is the same

Overall remarks

There are no results that prove the general thesis from abstract:  Bacteria can better attach to the positive charged ore surface at pH lower than 3. Nowhere any counts or pictures of bacteria on ore surface. Support your thesis by defining the surface charge or Zeta potential of bacterias at different pH. The pH in experiments with ore is higher than 3. That means the ore is negatively charged. Why do you believe that the bacteria attach to the mineral surface? Proof it by experimental data. The novelty will rise if you proof that the adaptation changes the Zeta potential of microbial cells and therefor the attachment bevavior.

Author Response

Thanks for your comment.

Reviewer 2 Report

Bioleaching of pyrrhotite with bacterial adaptation and biological oxidation for iron recovery

Bong-Ju Kim, Yong-Kwon Koh, Jang-Soon Kwon

 My comments:

1.

16 bioleaching under different initial pH conditions (2.8 and 3.2). Negatively charged bacteria could be

Should be initial           pH (pHini) subscript

 18 PZC (3.0). Under bacteria-adapted conditions, the leaching concentration of Fe (44.2 mg/L) at pHini

 pHini 19 of 2.8 was 2.1 times greater than that (21.3 mg/L) at pHini  pHini of 3.2

2.

33 ….. Currently, the bi-

34 oleaching method is attracting interest as an alternative method due to its environmental 35 friendliness and cost effectiveness.

 note / reference to a word currently-

the method has been known since the dawn of metallurgy:

“The Rio Tinto mines in South-Western Spain are usually considered as the cradle of biohydrometallurgy. These mines have been exploited since pre-Roman times for their copper, gold, and silver values. However, with respect to commercial bioleaching operations on an industrial scale, biohydrometallurgical techniques were introduced to the Tharsis mine in Spain earlier.”

  1.  

 70 oxidation on bioleaching efficiency at different pHs using (non-) adapted indigenous bac

71 teria.

non-adapted and adapted  or   non   and adapted

  1.  

94 the indigenous microbes existing in mine drainage (pH: 4.62 and Eh: 365 mV) located at

 Eh (Redox potential)

  1.  

 95 which was 82 m under the ground at the Ul-Jin Mine cave, and The microbes were cul

The

  1.  

135 where E is the leaching concentration at time t, EI is the maximum leaching concen

 Esubscript

 141 Jin Fe mine contained 5670 mg/kg of Fe, 140 mg/kg of Pb, 479 mg/kg of Cu, and 238 mg/kg 142 of Zn.

contained 5670 mg of  Fe /kg   of mine waste

   contained 5670 mg Fe /kg  mine waste

7.

4.Conclusions

291 In order to enhance the recovery of valuable metals from mine waste, various meth2

92 ods of bioleaching characterized by bio-adaptation and bio-oxidation were carried out us

293 ing a pyrrhotite sample obtained from an abandoned mine and indigenous bacteria (A.

294 ferrooxidans) inhabiting the acid mine drainage. The indigenous bacteria were cultured

295 and adapted for iron resistance. The pyrrhotite sample used here was prepared by crush

296 ing, grinding, sieving, and magnetic separation, and its pHpzc was estimated to be 3.0. Ac

297 cordingly, the initial pH of the bioleaching solution was controlled at 2.8 and 3.2 in view

298 of the surface charge of the pyrrhotite sample.

Most of the conclusions do not concern the results of Fe bioleaching from waste. Rather, it repeats the sentences from Chapter 2 (Materials and methods). Please briefly present the main results of the work, which are contained in subchapters 3 (Results and Discussion).

Author Response

Thanks for your comment.

Reviewer 3 Report

The article is useful for bioleaching of pyrrhotite.

Please check the following parts.

Line 30;   Fe is usually main sources from oxide, not from sulfides.

Line77;    20 mesh is 833 micron in Tyler and 841 micron ASTM, not 1130nmicron.

Line 79; Unit should be unified. 2000 Gauss is 0.2T like in Figure 2.

Line 135;  l of El is subscript.

In Figure 3, is it possible to write the error bars clearly?

In Figure 4, the color of graph is not clear. Please change the expression.

The characters in Figure is too small. Please write larger.

After the reaction has Jarosite not been observed?

Author Response

Thanks for your comment.

Round 2

Reviewer 1 Report

Dear authors,

thanks for beginning to improve the paper. I have some more remarks to the attachment of bacteria to mineral. The figure 7 was added to prove the better attachment of bacteria at lower pH values. Both parts of figure 7 differ in magnitude and support material. Fig 7a looks like a section of mineral and fig 7b looks like membrane filter. None of the pictures show slime layer. With the Zetasizer Nano-ZS  MPT-2 the zetapotential of cells could be measured and there were less speculation about the nature of interaction between mineral and bacteria.

Author Response

Thank you for the comment.

Round 3

Reviewer 1 Report

I understood your explainations. But in generell it would be nice to see a proof of postualtes in results as well.

Author Response

Thank you very much for your interest and advice on our paper.
